# Fructose and Mannose in Inborn Errors of Metabolism and Cancer

**DOI:** 10.3390/metabo11080479

**Published:** 2021-07-25

**Authors:** Elizabeth L. Lieu, Neil Kelekar, Pratibha Bhalla, Jiyeon Kim

**Affiliations:** Department of Biochemistry and Molecular Genetics, University of Illinois, Chicago, IL 60607, USA; elieu3@uic.edu (E.L.L.); nkelek2@illinois.edu (N.K.); pbhall4@uic.edu (P.B.)

**Keywords:** fructose and mannose, inborn errors of metabolism, cancer

## Abstract

History suggests that tasteful properties of sugar have been domesticated as far back as 8000 BCE. With origins in New Guinea, the cultivation of sugar quickly spread over centuries of conquest and trade. The product, which quickly integrated into common foods and onto kitchen tables, is sucrose, which is made up of glucose and fructose dimers. While sugar is commonly associated with flavor, there is a myriad of biochemical properties that explain how sugars as biological molecules function in physiological contexts. Substantial research and reviews have been done on the role of glucose in disease. This review aims to describe the role of its isomers, fructose and mannose, in the context of inborn errors of metabolism and other metabolic diseases, such as cancer. While structurally similar, fructose and mannose give rise to very differing biochemical properties and understanding these differences will guide the development of more effective therapies for metabolic disease. We will discuss pathophysiology linked to perturbations in fructose and mannose metabolism, diagnostic tools, and treatment options of the diseases.

## 1. Introduction

The cultivation of grains to create a reliable source of carbohydrates was a crucial step in the cultural transition from nomadism to sedentism. Humans have evolved enhanced utilization of glucose as the central carbon source that feeds into catabolic and anabolic pathways, including the long-term storage of glucose as glycogen. Glucose, the major dietary monosaccharide with six carbon atoms (hexose), is an essential part of a healthy diet, and as such, a great emphasis has been placed on the study of glucose metabolism. Through a diversity of mechanisms and interactions, dysregulations of glucose metabolism—namely, perturbations in central carbon metabolism (glycolysis, pentose phosphate pathway (PPP), and tricarboxylic acid (TCA) cycle)—are recognized as key steps not only in metabolic disorders (e.g., obesity, insulin intolerance, nonalcoholic fatty liver disease) but also in cancer progression [1,2,3]. However, glucose is not the only hexose metabolized by the cell; fructose and mannose encompass two important hexoses that cells use for energy production [4] and intricate cellular processes, such as glycosylation [5]. While fructose and mannose are not essential to the human diet, the prevalent role of metabolism in disease has shifted the field to place an increased importance on the study of these two non-essential hexoses in recent years. 

Consumed in lower amounts as recently as a century ago, fructose, oftentimes called fruit sugar, has been extensively studied since its discovery by Augustin-Pierre Dubrunfaut in 1847 and the elucidation of its configuration by Emil Fischer’s stereochemistry study in the late 1800s. Fructose has become increasingly present in the Western pattern diet through the increased availability of sucrose and high-fructose corn syrup. As fructose has transitioned to a stalwart part of the average diet in the forms of processed breads and colas, fructose has been shown to play a chronologically and statistically significant role in the increased prevalence of obesity and metabolic syndrome, particularly in the United States [6,7]. Given that the diagnostic criteria for metabolic syndrome include central obesity, hyperglycemia, dyslipidemia, and hypertension, fructose’s relationship with the development of high morbidity conditions emphasizes the importance of studying and understanding fructose metabolism as it relates to disease development and progression. While enzymatic function in fructose metabolism can correspond to poor metabolic prognoses, so too can enzymatic deficiencies in the pathway. To date, three inborn errors are known in the pathway of fructose metabolism: fructokinase deficiency, aldolase B deficiency, and fructose-1,6-bisphosphatase deficiency (Figure 1). Together, these deficiencies emphasize the importance of fructose metabolism through anabolic and catabolic processes: the trapping of fructose in the cell, the contribution of fructose to metabolic intermediates, and the roles of fructose in glycogenesis. 

Mannose, by contrast to fructose, is more discrete dietarily. However, mannose metabolism is fundamental to cellular health. Mannose plays an important structural role as mannans, hemicellulose, and cellulose [8] in a variety of beans, fruits, and plants. At the cellular level, D-mannose and its derivatives are key metabolites for most glycosylation reactions. Fascinating is the etymology of mannose—the name mannose is derived from the biblical word manna, the pivotal food source that sustained the Israelites during their arduous journey in the Sinai Peninsula [8]. This profound name seems fitting, given the critical contribution of mannose to cells through glycosylation. First epimerized from glucose by utilizing the Lobry de Bruyn–Van Ekenstein transformation in the late 1890s, mannose’s structural similarities to glucose make mannose biochemically interesting to study, even more so based on glucose’s role in central carbon metabolism. Mannose is present in relatively large quantities in the dry weight of spent coffee grounds [9] while also present in the dry weight of litchi pericarp [10]. Mannose can also be dietarily sourced through the consumption of egg whites via ovalbumin and ovomucoid [11], two main glycoproteins with high mannose content. Although mannose’s role in normal physiology and pathophysiology are not completely understood at present, a few inborn errors of mannose metabolism have been studied [12]. Inborn errors of mannose metabolism discussed in the review are congenital disorders of glycosylation (CDGs) caused by loss of function of three enzymes in the mannose metabolism pathway: mannose phosphate isomerase (MPI, MPI-CDG), phosphomannomutase 2 (PMM2, PMM2-CDG), and ALG11 alpha-1,2-mannosyltransferase (ALG11, ALG11-CDG) (Figure 1). All three CDGs exhibit defects in N-glycosylation, a functional consequence of mannose metabolism. Interestingly, mannose appears to have opposite effects on tumor growth by comparison to fructose. Millimolar concentrations of mannose supplement slow mouse subcutaneous tumor growth. This occurs as the intracellular accumulation of mannose-6-phosphate (M6P) impairs the further processing of glucose in glycolysis, the TCA cycle, the PPP, and glycan synthesis, implying that dietary mannose could be a safe therapeutic intervention in cancer treatment.

The ensuing review will discuss the mechanisms of inborn errors of fructose and mannose metabolism, and how these diseases are diagnosed and treated. The review will also explore the link between dysfunction of fructose and mannose metabolism and other metabolic disorders, such as cancers.

## 2. Fructose Metabolism Disorders

### 2.1. Inborn Errors of Fructose Metabolism

#### 2.1.1. Essential Fructosuria

Metabolism of fructose and the resulting energy realization is achieved through a process known as fructolysis. The first step of fructolysis after fructose enters the cells is the phosphorylation of fructose to fructose-1-phosphate (F1P) by fructokinase (Figure 1). Fructokinase, also known as ketohexokinase (KHK), is an ATP-dependent enzyme that converts fructose to F1P. Alternative splicing of the *KHK* gene located on chromosome 2 gives rise to two isoforms of fructokinase: fructokinase A (KHK-A) and fructokinase C (KHK-C). KHK-C is the isoform predominantly expressed in the liver, kidney, and intestine while KHK-A is more ubiquitously expressed [13]. While KHK-C has the dominant role in liver fructose metabolism, whole-body deletion of both isoforms has been found to be even more preventative against metabolic syndrome, suggesting an important role for KHK-A in fructose metabolism in extrahepatic tissues with low expression of KHK-C [14]. Small amounts of fructose are usually metabolized by the intestine, but if the levels of fructose exceed the capacity of intestinal fructose absorbance, fructose can be metabolized by the liver and colon microbiota [15]. In the absence of KHK, however, fructose cannot be further metabolized even in the liver. It has been shown that the fate of unmetabolized fructose diverges into skeletal muscle or adipose tissue, where it can be metabolized by hexokinase instead, or excretion into urine [16,17]. KHK deficiency overall leads to fructosuria through the accumulation of fructose in the blood and eventual excretion of unmetabolized fructose into the urine, hence the name fructosuria. Mouse models have been used to study the role of Khk in the liver, renal, and intestinal contexts [16,18]. While Khk has an important role in modulating sugar intake in all three organs, metabolic syndrome is specifically driven by liver, not renal or intestinal, Khk. Deletion of liver Khk leads to complete protection against the development of sugar-induced metabolic syndrome [18] (Figure 2). The study showed a further tissue-specific role of Khk. Intestinal Khk, but not liver Khk, is involved in potentiating the sweet taste preference, thus promoting sugar intake [18]. Other mouse studies with Khk suppression demonstrated additional protective roles of Khk deficiency. Fructose consumption could still lead to obesity in Khk KO mice, but the mice displayed no sign of hepatic inflammation on a high-fat diet when compared to a low-fat diet [19]. Pharmacological inhibition of Khk could reduce the adverse effects of acute kidney injury driven by endogenous fructose production [20]. Collectively, these studies suggest potentially protective roles of KHK suppression in metabolic syndrome [10,11,12,13,14,15,16,17,18,19,20].

KHK deficiency is a diagnosed inborn error of fructose metabolism in human patients (Table 1). First described in 1876 by Zimmer and Czapek [21], hepatic fructokinase deficiency is an autosomal recessive disorder. Affected patients are diagnosed with essential fructosuria, characterized by elevated concentrations of fructose in the blood and urine [13,22] (Table 1). Upon fructose ingestion, 10–20% of the administered fructose dose is excreted in patients compared to 1–2% excretion for normal subjects [21]. Unlike other metabolic diseases, however, the incidence of essential fructosuria is difficult to gauge, considering the disease has unharmful symptoms and hence may be underdiagnosed. Nevertheless, there are a few methods that have been used to identify KHK deficiency: enzyme assay on liver biopsies [21] and fructose loading followed by ^31^P-NMR studies [23], an approach that is used to measure ATP, phosphomonoesters, and inorganic phosphate concentrations. Fructokinase activity consumes ATP to produce F1P; therefore, patients exhibiting unchanged ATP levels following administration of fructose have inactive fructokinase. Since the disease does not have health-threatening clinical presentation, there is no treatment necessary for essential fructosuria. It is understood that loss of fructokinase activity in essential fructosuria does not present a health threat, so research efforts have been directed toward understanding how fructokinase activity contributes to other physiological processes. 

#### 2.1.2. Hereditary Fructose Intolerance

Fructose metabolism continues, with F1P being further metabolized in a reversible reaction by the enzyme aldolase B (ALDOB), encoded by the *ALDOB* gene on chromosome 9, into dihydroxyacetone phosphate (DHAP) and glyceraldehyde (GA) (Figure 1). Functional ALDOB enzymatic activity is important for fructose metabolism because DHAP and GA can be further converted into triglycerides and pyruvate through glycolysis. These glycolytic products sustain cell growth by providing cellular building blocks and intermediates for energy production. Expressed in the liver, small intestine, and proximal renal tubule, ALDOB prevents accumulation of F1P. ALDOB deficiency or malfunction leads to elevated F1P levels, and disrupts pathways related to fructose metabolism, including glycolysis, gluconeogenesis, fatty acid synthesis, and glycosylation. Therefore, the health-threatening consequences, if perturbed fructose metabolism is not corrected, include hypoglycemia, metabolic acidosis, and deficient glycosylation, all contributing to liver and kidney damage [24,25] (Table 1). 

Metabolic acidosis, characterized by increased hydrogen ion levels in plasma, is caused by multiple factors related to F1P accumulation. One of these factors is F1P allosteric activation of pyruvate kinase leading to the accumulation of TCA cycle precursors, including lactate, pyruvate, and alanine [26]. The role of F1P in gluconeogenesis also contributes to metabolic acidosis. F1P is a competitive inhibitor of glycogen phosphorylase, which mediates glycogen to glucose conversion in the liver and kidney. This blockade causes accumulation of acidic metabolites, including lactate, which would usually be consumed in the process of gluconeogenesis [24,27]. Another altered pathway affected by F1P accumulation is fatty acid metabolism. Elevated F1P and subsequent activation of adenosine monophosphate (AMP) deaminase followed by inhibition of AMP-activated kinase (AMPK) leads to reduced β-oxidation of fatty acids and accumulation of triglycerides in the liver [28,29]. Furthermore, F1P also impacts glycosylation since it is a competitive inhibitor of liver MPI, which is critical for the N-glycosylation pathway [30]. 

The disease characterized by ALDOB deficiency is hereditary fructose intolerance (HFI), an autosomal recessive inborn error of metabolism with an incidence of approximately 1 in 20,000 newborns [31,32,33]. As the name suggests, HFI prevents dietarily consumed fructose from being metabolized within the body. Due to F1P’s inhibitory effect on N-glycosylation [30], HFI is not only recognized as an inborn error of fructose metabolism, but is also considered a secondary CDG. HFI was originally characterized in 1956 by Chambers and Fratt as “idiosyncrasy to fructose” after observing the development of nausea, vomiting, abdominal pain, and faintness in patients after consuming fructose. Over the next four to five years, the pathophysiology was delineated when the disease was pinpointed to ALDOB defects [34,35]. Since then, seven common mutations of *ALDOB*, which comprise 82% of mutant alleles worldwide and lead to HFI, have been identified: A149P, A174D, N334K, Δ4E4, R59OP, A337V, and L256P [36]. While common mutations have been identified and are frequently used for genetic screening of HFI as a diagnostic tool, within the American population of individuals with *ALDOB* deficiency, over 33% of mutant HFI alleles are reported as unknown [37], leaving research to be done on identifying the disease-causing mutations.

Since HFI is considered to be an inborn error of metabolism, symptoms of the ailment manifest early on and are triggered by fructose consumption because defective ALDOB does not allow further processing of F1P. Notably, symptoms can often be detected upon dietary changes, especially those occurring during infancy when transitioning from breastmilk to formula or baby food [34,38] (Table 1). These symptoms arise because breast milk contains lactose, which is comprised of glucose and galactose subunits, whereas select infant milk formulas and baby foods often contain sugar in the form of sucrose, which is a glucose and fructose disaccharide [39], therefore providing a source of fructose for F1P accumulation. While presentation of the disease takes many forms, some of the most common symptoms in infants include difficulty feeding, vomiting, failure to thrive [38], and aversion to sweet foods [40,41], which are related to the physiological consequences of F1P accumulation.

Both hepatic and renal function are affected by ALDOB deficiency. Because of increased accumulation of triglycerides in the liver, HFI patients show a higher prevalence of fatty liver not linked to obesity or insulin resistance [29]. Pertaining to kidney function, rapid ATP consumption by renal cells to mitigate the lack of downstream metabolites produced by ALDOB leads to depletion of inorganic phosphate and ATP [25,42]. This ATP reduction causes increased uric acid production, magnesium release, impaired protein synthesis, and ultrastructural impairment, all contributing to organ damage [43]. Additional markers of HFI include elevated levels of fructose or amino acids in urine caused by organ damage and an inability to synthesize proteins necessary for proper function [25,44,45]. Furthermore, metabolites related to methionine metabolism serve as markers of HFI. For example, elevated S-adenosylmethionine/S-adenosyl-L-homocysteine (SAM/SAH, two intermediary metabolites in the methionine cycle) ratios are typically identified in patients with HFI [29], suggesting an altered methionine cycle, which impacts gene expression through epigenetic regulation (histone and/or DNA methylation). 

As fructose consumption leads to the accumulation of F1P in HFI patients, strict dietary changes related to sugar intake are an essential part of managing HFI. Additional sources of fructose include sorbitol (sugar alcohol) and sucrose (disaccharide: fructose-glucose), sugars that can be converted to fructose. Thus, dietary restrictions often require avoidance of foods containing fructose, sucrose, and sorbitol, which can be commonly found in fruits, processed foods, and artificial sweeteners (Table 1). While it is unfeasible to completely restrict sources of fructose from one’s diet, studies have identified markers that provide an idea of what threshold of fructose can be consumed before adverse effects are experienced. For instance, serum carbohydrate deficient transferrin (CDT) may be indicative of fructose/sorbitol/sucrose intake [41]. To combat damage caused by fatty liver disease related to HFI, studies have shown that dietary methionine restriction can increase β-oxidation, which is reduced in HFI patients, by regulating SAM/SAH ratios affecting the methionine cycle [29,45]. Studies have also investigated the long-term effects of fructose/sorbitol/sucrose-free diets on disease management. It was found that even with a restrictive diet, most patients showed mild signs of liver disease; however, this liver disease was not complemented by damage progression. HFI patients in the same study also exhibited elevated systolic blood pressure, epidermal growth factor receptor (eGFR), and plasma sE-selectin levels compared to healthy controls [46]. Therefore, when treating HFI patients, the effects of fructose restriction on blood pressure and renal function should also be considered.

Besides dietary restrictions, researchers have considered other treatments against HFI. A mouse model study suggested that the absence or inhibition of the upstream enzyme KHK is sufficient to combat the hypoglycemia and liver and intestinal injury associated with HFI. Indeed, they found that *Khk^−/−^*/*Aldob^−/−^* mutant mice have reduced HFI-associated hepatic inflammation and fibrosis in comparison to solely *Aldob^−/−^* mutated mice [47]. Therefore, a possible therapeutic drug treatment inhibiting KHK may improve the conditions of patients suffering from HFI. Overall, although there may be severe consequences concerning an inborn error of *ALDOB*, current treatments and therapies have shown promise to lessen the severity of symptoms and combat HFI.

#### 2.1.3. FBPase Deficiency

Inborn errors of fructose metabolism are not limited to dysfunction of enzymes in direct fructose metabolic pathways. Fructose 1,6-bisphosphatase (FBPase) or FBP1, a gluconeogenic enzyme, is also associated with an inborn error of fructose metabolism. After F1P is catabolized to DHAP and GA by ALDOB, the fructose-derived metabolic intermediates are further metabolized in the same way to glucose-derived metabolites by entering the glycolytic pathway. FBP1, encoded by the *FBP1* gene on chromosome 9, catalyzes the hydrolysis of fructose 1,6-bisphosphate (F1,6BP) to F6P in the presence of divalent cations, acting as a rate-limiting enzyme in gluconeogenesis and potentially antagonizing glycolytic flux (Figure 1). It is expressed in several tissues, with high levels of activity within the kidney and liver and contributes significantly to glucose production. Studies have found multiple isozymes with FBPase activity in humans, including the muscle and liver isoform. However, the muscle isoform (FBP2) has different kinetic characteristics in comparison to the liver isoform (FBP1) and has been found not to be affected in patients with FBPase deficiency. Therefore, the disease FBPase deficiency in this section refers to the deficiency of FBP1.

Mutations in FBP1 cause FBPase deficiency, a rare autosomal recessive disorder (Table 1). The disease was first described in 1970 by Baker and Winegrad. They noted frequent episodes of hypoglycemia and lactic acidosis in a pair of siblings and later discovered that the symptoms were due to FBPase deficiency. FBPase deficiency prevents glucose production through gluconeogenesis. Thus, F1,6BP cannot be used to generate glucose to correct hypoglycemia [48]. Prolonged hypoglycemia can cause defects in normal growth and psychomotor development and can even be lethal [49]. Further, the blockade of gluconeogenesis by FBPase deficiency leads to the accumulation of gluconeogenic substrates lactate, pyruvate, glycerol, and alanine [50,51]. An inability of patients with FBPase deficiency to convert lactate into glucose results in lactic acidosis (Table 1). Additionally, the accumulation of pyruvate can increase malonyl-CoA synthesis, which prevents the entry of long-chain fatty acyl-CoA into the mitochondria. This causes a reduction in ketogenesis as well as an accumulation of fatty acids in the liver and plasma [52]. Other symptoms affiliated with the disorder include hepatomegaly and episodic acute crises of hyperventilation, coma, and seizures due to acidoketosis as well as lactic acidosis [49,51,53], which occur upon triggers activating catabolic pathways (i.e., fasting, infections, and stress) (Table 1). Later episodes of FBPase deficiency can include the aforementioned symptoms in addition to irritability, drowsiness, dyspnea, tachycardia, and muscular hypotonia, all of which can be triggered by febrile infectious disease.

The prevalence of FBPase deficiency is less than that of HFI, as there is an approximate incidence of 1:350,000 in the Dutch population and <1:900,000 in the French population [54]. Multiple mutation sites have been identified across different geographical populations, and it is generally found that consanguinity increases the risk of homozygous mutation of *FBP1* [49]. Of the 22 different mutations of *FBP1* reported, no single mutation is particularly frequent except the c.961 insertion guanine mutation, which has been reported to be responsible for 46% of the mutated alleles in Japan [52]. Interestingly, not all mutations causing FBP1 deficiency are within the coding region. Studies have shown that several FBPase-deficient patients have mutations outside of the *FBP1* coding sequence. This suggests that mutations within the promoter region or in genes that affect regulation of FBPase activity may also cause FBPase deficiency. Mutation of *FBP2* encoding fructose-2,6-bisphosphatase, the main physiological regulator of FBPase, has also been associated with FBPase deficiency [52].

To identify whether patients have these genetic mutations and to explain their clinical presentation, several methods have been used to diagnose patients with FBPase deficiency. For one, fructose tolerance tests have been used as a diagnostic tool, similarly to HFI. However, adverse neurological effects and the advent of new diagnostic methods have caused this method to be discontinued [55]. Currently, molecular analysis on peripheral leukocyte DNA is being performed [52]. If a mutation is not identified, however, clinical and laboratory findings may suggest a diagnosis. For instance, a liver biopsy and subsequent enzymatic activity assay may be done to diagnose patients with FBPase deficiency. Furthermore, tissue FBP1 enzymatic activity may be directly assessed by spectrophotometry to measure NADPH formation in the presence or absence of AMP, a specific inhibitor for FBP1 [48,55]. Additional biomarkers of the disorder include elevated glycerol 3-phosphate, lactate, and ketone levels in urine when an individual is in a diseased state compared to a healthy state [53,56,57]. Despite multiple markers for identifying FBPase deficiency, there are concerns about misdiagnosis because other energetic defects or glycogen storage diseases are often suspected before considering this rare disease a possibility. This is because they share similar symptoms, including an enlarged liver and hypoglycemia that triggers the episodes characteristic of FBPase deficiency [48,53]. In those cases where multiple diseases diagnoses might be suggested based on clinical presentations, further lab work is conducted to narrow down the possibilities. For example, patients with FBPase deficiency display normal levels of hepatic aldolase activity, eliminating hereditary fructose intolerance as a potential diagnosis. Such distinctions prevent frequent misdiagnoses from occurring in a clinical setting.

Although severe symptoms manifest during provoked episodes, the prognosis is favorable once diagnosis is established and treatment begins. Since FBPase-deficient patients often present with hypoglycemia, intravenous (IV) and oral glucose should be administered to keep blood sugar levels elevated (Table 1). Treatment of the disorder requires limitation of fructose intake and prevention of prolonged fasting, which prompts gluconeogenesis [54]. Future episodes are easily prevented by avoiding fasting and ensuring proper nutrition intake during infections. Certain fruits, such as apples, pears, grapes, and cherries, have a higher ratio of fructose to glucose, which should be avoided in cases of FBPase deficiency [53]. It has also been shown that providing uncooked cornstarch mixed with water at midnight prevents nocturnal hypoglycemia, improving the overall clinical outcome [53] (Table 1). Following glucose administration, patients’ blood pH may remain below <7.1, which can be corrected by the addition of sodium bicarbonate; however, this is not routine since acidosis is usually rapidly corrected [51]. Most importantly, proper diagnosis during the early stages of the disease is necessary to prevent treatment for an incorrect diagnosis [48,58].

### 2.2. Fructose Metabolism in Cancer

Cancer cells often rewire their metabolism to sustain survival and growth. The procurement of nutrients is critical for meeting biosynthetic and bioenergetic demands and protecting against reactive oxygen species formed during proliferation and metastasis. Some of these metabolic changes involve the increased consumption of fructose in low glucose conditions or the regulation of enzymes to support alternative metabolic pathways. There have been comprehensive studies and reviews documenting the role of fructose in cancer metabolism [59,60,61,62]; therefore, we will briefly focus on some recent findings related to fructose metabolism through central carbon pathways. Since the consumption of fructose has increased over the past decades, understanding how cancers metabolize fructose has become more critical and encourages a consideration of dietary changes or alternative therapeutic strategies that might target fructose-related pathways to mitigate cancer progression.

The cancer microenvironment affects how cancer cells utilize certain nutrients to favor their growth. For instance, in glucose-depleted conditions, cancer cells can rely on fructose as an alternative resource. One way of acquiring fructose from the environment is through the fructose transporter, GLUT5, which is upregulated in acute myeloid leukemia and lung adenocarcinoma to compensate for low glucose conditions [63,64] (Figure 3). Even the nutrient composition of the tumor microenvironment can drive cancer development.

For example, a study using breast cancer cells revealed how cancer cells cultured in normal media or media where glucose was replaced with fructose generated different phenotypes [65]. These differences included changes in cell surface glycan structures, genes related to glycan assembly, cellular structure, adhesion, and invasion. Furthermore, exposing less aggressive cancer cells to fructose-rich conditions led to enhanced migration and invasive potential [65]. These changes can be attributed to the effect of fructose on altered glycosylation structures, which have been previously linked to invasive and metastatic phenotypes [66]. 

As fructose is easily acquired through diet and it has been shown that the nutrients in the tumor microenvironment affect cancer progression, it is important to mention how fructose consumption has been studied in the cancer context. Like glucose, fructose can be metabolized to support glycolysis and fatty acid synthesis. High-fructose corn syrup (HFCS) consumption has been reported to promote the growth of intestinal tumors. By increasing fructose and glucose concentrations in the intestinal lumen and serum, HFCS potentiates fructose utilization of intestinal tumors; tumors can directly transport and catabolize fructose, specifically with the activity of KHK, to generate F1P from fructose and accelerate glycolysis and fatty acid synthesis. Fructose-rich conditions can also rewire other metabolic pathways, a phenomenon observed in acute myeloid leukemia, where fructose-rich conditions cause an upregulation of the de novo serine synthesis pathway to support α-ketoglutarate generation for consumption in the TCA cycle [67] (Figure 3). Therefore, fructose acquisition through one’s diet can greatly impact cancer development.

In addition to dietary restriction, many studies have investigated enzymatic alterations of fructose metabolism in cancer. While the inborn error of KHK deficiency was described as not causing adverse effects, it has been found that in breast cancer, fructose intake induces KHK-A (ubiquitously expressed KHK isoform) entrance into the nucleus, leading to a signaling cascade triggering the metastatic phenotype [68].

Alterations in ALDOB activity also impact cancer aggressiveness. In colorectal cancer, metastatic cancer cells upregulate ALDOB in the liver to fuel central carbon metabolism as well as gluconeogenesis. Importantly, dietary restriction of fructose reduced liver metastases, leading to longer survival in mice [69]. Alternatively, given the heterogeneous nature of cancer, aggressive hepatocellular carcinoma is associated with ALDOB downregulation, and stable expression of ALDOB reduced cell migration and lung and intrahepatic metastases [70] (Figure 3).

The gluconeogenic enzyme FBP1 has been studied in the context of cancer. FBP1 has been identified as having a tumor suppressive effect in prostate cancer, gastric cancer, and clear cell renal cell carcinoma (ccRCC) (Figure 3). The mechanism in prostate cancer involves loss of FBP1 expression, which causes activation of the mitogen-activated protein kinase (MAPK) pathway to elicit cancer cell invasion and metastasis [71]. Additionally, the inhibitory epithelial-to-mesenchymal transition (EMT) effect of FBP1 has been identified in gastric cancer, where loss of FBP1 expression supported cancer cell growth through increased glycolysis [72]. In ccRCC, FBP1 has a similar effect, where loss of FBP1 is associated with poorer prognosis. This phenomenon has two explanations: FBP1 expression reduces glycolysis, affecting PPP flux, which is required for nucleotide biosynthesis and management of oxidative stress. FBP1 also exhibits its enzymatic function-independent effects though its role in inhibiting the hypoxia inducible factor (HIF) in the nucleus, which subsequently disrupts glucose metabolism, effectively preventing aerobic glycolysis [73].

The involvement of fructose in various metabolic pathways allows cancer cells to rewire their metabolism to catabolize fructose; however, this overreliance on fructose creates opportunities for the development of drugs targeting fructose metabolism. While the loss of fructose-related metabolic enzymes in inborn errors of disease can have detrimental outcomes, the targeting of some of the same enzymes in certain cancer settings can have beneficial effects. Therefore, it is essential to understand the diverse role of fructose across disease contexts in order to optimize how to treat those diseases.

## 3. Mannose Metabolism Disorders

### 3.1. Inborn Errors of Mannose Metabolism

Mannose metabolism is intricately linked to glucose as well as fructose metabolism through isomerization reactions. F6P, via the MPI reaction, serves as a bridge point between glucose metabolism (central carbon metabolism and the hexosamine biosynthesis pathway) and mannose metabolism (glycosylation) (Figure 1). Mannose metabolism is critical for N-glycosylation, the most abundant and critical post-translational modification. Thus, inborn errors of mannose metabolism are ascribed to the congenital disorders of glycosylation (CDG). It was originally named carbohydrate-deficient glycoprotein syndrome by Jaeken. In 1980, he first studied the blood iron of two consanguineous infants and linked reduced iron transport in the blood in patients to altered glycoprotein structure [74]. Jaeken devised two subtypes of CDGs based on specific alteration patterns in glycoprotein structure: CDG-type I and CDG-type II. CDG-type I disorders describe deficiencies in the formation of lipid-linked oligosaccharides and their transfer into the lumen of the ER, whereas CDG-type II disorders are defects in the processing of the lipid-linked oligosaccharides once bound to protein. In 2009, however, Jaeken advocated for renaming the disorders based on the enzyme deficiency causing the disorder, i.e., CDG-Ia was renamed as PMM2-CDG, CDG-1b as MPI-CDG, and CDG-IIa as MGAT2-CDG [75]. While there are existing reviews that extensively cover CDGs in depth [76,77,78], this review is particularly interested in covering three type I CDGs: MPI-CDG, PMM2-CDG, and ALG11-CDG (Figure 1 and Table 1).

#### 3.1.1. MPI-CDG

MPI, encoded by the *MPI* gene on chromosome 15, catalyzes the interconversion of F6P, a ketohexose, to M6P, an aldohexose. MPI plays a critical role in maintaining the supply of D-mannose derivatives because most M6P originates from F6P by MPI-mediated interconversion, rather than from extracellular mannose [79] (Figure 1). M6P is further metabolized to its derivatives, GDP-mannose and GDP-fucose. GDP-mannose is the key nucleotide sugar to initiate N-glycosylation, whereas GDP-fucose is a part of N-linked glycan core structures as well as the Lewis^x^ and sialyl Lewis^x^ antigens. N-linked glycosylation is critical for ER homeostasis because protein-linked oligosaccharides are used as recognition and timing markers for glycoprotein quality control pathways that discriminate between correctly folded proteins and misfolded proteins destined for ER-associated degradation.

With (1) N-glycosylation being one of the most abundant post-transitional modifications, and (2) the essential role of MPI in producing D-mannose derivatives [80], it is not surprising that complete loss of MPI causes embryonic lethality [81,82]. Thus, human inborn errors linked to MPI are not through the enzyme’s complete loss, but rather insults to the enzyme’s efficacy for the bidirectional isomerization of F6P to M6P. MPI-CDG, a disorder characterized by MPI deficiency formerly known as CDG-Ib, is heterogeneously passed in an autosomal recessive fashion [83]. First studied in young Canadian patients with profound gastrointestinal symptoms by Pelletier in 1986 [84], it was not until Jaeken’s 1998 retrospective study that Pelletier’s novel diagnosis was correctly marked as MPI-CDG [85]. The disease exhibits severe disruption of gross glycosylation processes, arising in patients through coagulopathies and failure to thrive [86,87]. The symptoms result from the altered activation of enzymes in the production of the lipid-linked oligosaccharide core, increased ER stress following protein misfolding, or from changes in the activity of glycosidases and glycotransferases (Table 1). Interestingly, MPI-CDG symptoms are atypical by comparison to other disorders in the CDG family of disorders, which are described later, and show a marked lack of neurological pathophysiology [80,83,88]. MPI-CDG patients, of which 35 have been described as of 2020, present with an established triad of hepatic, gastroenterological, and endocrine symptoms [89,90] (Table 1).

This hallmark three-pronged pathophysiology can serve as a useful tool to pathologists in differentially diagnosing MPI-CDG from other CDGs. Some symptoms that MPI-CDG patients experience include liver fibrosis, vomiting, hyperglycemia, protein-loss enteropathy, and failure to thrive [83,91,92]. Diagnosis of MPI-CDG, as well as other type I and II CDGs, relies on transferrin isoelectric focusing (TIEF) to detect defective N-glycosylation [93,94,95]. The glycoprotein transferrin is found in abundance in the blood and facilitates iron transport. Transferrin is post-translationally glycosylated at multiple residues and each N-linked glycoprotein antenna carries two terminal *N*-Acetylneuraminic acids, the predominant sialic acid found in human cells. The presence of these sialic acids, or lack thereof, can allow the separation of transferrin isoforms by charge, allowing clinicians to analyze and quantify glycosylation in patients, and previously classify type I or type II CDGs. Alongside TIEF screening methods, MPI-CDG is being increasingly examined through high-performance liquid chromatography (HPLC) and capillary zone electrophoresis [96,97]. Studying adult populations of MPI-CDG patients is important in growing our understanding of the long-term attritional effects of hypoglycosylation. Additionally, MPI-CDG is thought to be underdiagnosed in adult populations given the disorder’s symptomatic relatedness to excessive alcohol consumption [98]. Despite existing efficacious diagnostic tools, there has been a greater push towards using robust genomic analysis strategies, such as multigene panels, on symptomatic MPI-CDG patients to better understand the underlying pathophysiology of insulted N-glycosylation [95].

Interestingly, MPI-CDG is one of the few CDGs with a known treatment: patients have been shown to have relief of symptoms following daily mannose supplementation [80,99,100,101] (Table 1). However, it should be noted that congenital liver disease typically progresses in MPI-CDG patients despite mannose treatment [102]. For example, one group was able to correct all symptoms of an MPI-CDG patient that presented atypically with no hepatopathology through mannose treatment [101], whereas another group found that, despite mannose treatment, liver transplantation was necessitated due to the progressive nature of the patient’s liver fibrosis [102]. Understanding the molecular basis of persistent liver disease and avenues of treatment by or in conjunction with mannose supplementation will further the ability to treat MPI-CDG more holistically. These treatment regimens of mannose supplementation leverage serum uptake of mannose through GLUT and subsequent phosphorylation by hexokinase to bypass the deficient hexose isomerization, which then ultimately rescues glycosylation. Evident is the critical role of MPI in glycosylation and how disease can arise from MPI deficiency.

#### 3.1.2. PMM2-CDG

Downstream of MPI, phosphomannomutase 2 (PMM2), encoded on the *PMM2* gene on chromosome 16, catalyzes the isomerization of M6P to mannose-1-phosphate (M1P) (Figure 1). Subsequently M1P is utilized as a substrate to produce GDP-mannose, a critical mannose donor molecule in the biosynthesis of the core oligosaccharide in the N-glycosylation pathway. Inborn errors related to PMM2, deficiencies in PMM2 passed on through an autosomal recessive inheritance pattern, are known as PMM2-CDG, formerly CDG-Ia (Figure 1). Deficient PMM2 in patients with PMM2-CDG, as well as in mouse embryonic fibroblast models, causes hypoglycosylation due to defects in the N-glycosylation pathway. The inability to produce GDP-mannose limits the N-glycosylation pathway through decreased production of the lipid-linked oligosaccharide cores, which in turn affects protein folding, localization, and signaling.

PMM2-CDG was the CDG studied in Jaeken’s pioneering study in the 1980s, and the disorder, then CDG-Ia, was retroactively recognized by Jaeken in 1998 as being PMM2-CDG [85]. Being the most common CDG, PMM2-CDG is well studied in the clinic relative to other CDGs. By broadly impacting the efficacy of glycosylation, PMM2-CDG heavily impacts the nervous system through the disruption of extracellular matrix (ECM) anchorage in nerve myelination [103]. Furthermore, N-glycosylation has been implicated in neural signaling through the trafficking of dopamine receptor subunits to the membrane [104], adding to the essential roles of glycosylation in the nervous system (Table 1). Insults to the nervous system, such as poor anchorage of myelination to the ECM or reduced trafficking of neurotransmitter receptors to the surface of neurons, may broadly impact the central nervous system, and this could explain the high mortality rate for infant PMM2-CDG patients [105].

PMM2-CDG manifests through a diversity of mutations, with heterogeneity of frequent mutations in different populations. For example, among PMM2-CDG patients from Italy, the most common mutation is the L32R mutations [106], whereas the most common mutations for Scandinavian and French populations are R141H, D65Y, F119L, and E139K [107]. Such heterogeneity of mutations may explain the diversity of outcomes found in PMM2-CDG patients, with symptoms ranging from mild [108] to severe [109,110], among affected individuals. Studies from both human patients and mouse models have shown that PMM2-CDG appears to manifest uniquely by age group, leading to three distinct classifications: infantile multisystem [111,112], late-infantile and childhood ataxia-intellectual disability [113], and adult stable disability. During the infantile stage, patients show developmental delays, failure to thrive, and neuropathy. In the late infant stage, patients show ataxia, language and motor delays, and stroke-like episodes; in the adult stable stage, patients present with premature aging, abnormal sexual development, and increased risk of deep vein thrombosis [111,114,115,116,117].

Similar to MPI-CDG, diagnosis of PMM2-CDG is reliant on TIEF to uncover hypoglycosylation in patients exhibiting symptoms of PMM2-CDG [93,94]. On the horizon are approaches that leverage new facial scanning technologies to recognize dysmorphic features (such as upslanted eyes, wide mouth, and long philtrum) as biomarkers of suspected PMM2-CDG [118]. While mannose treatment has been shown to be successful in rescuing the hypoglycosylation and symptoms of PMM2-CDG both in vitro and in murine in vivo settings, the relief of symptoms is yet to be consistently shown in human clinical trials [119,120,121,122]. For example, a cohort study observing the potency of intravenous mannose infusion over a period of 5 months during the first year of life showed no biochemical or clinical improvement [119]. Given that treatment options utilizing mannose therapies lack consistent efficacy among patient populations, the molecule β-glucose-1,6-bisphosphate is being tested as a possible treatment for insulted PMM2 enzymatic activity in PMM2-CDG [123]. Utilizing better genomic screenings to uncover changes from consensus at specific CDG loci in children presenting with symptoms or early in life [95] will allow for not only a better understanding of the mechanisms underlying how PMM2 pathophysiology affects healthy development in PMM2-CDG patients, but also possibly elucidate efficacious therapeutic targets that may guide treatment for patients.

#### 3.1.3. ALG11-CDG

Further downstream of the two aforementioned enzymes (MPI and PMM2) is GDP-Man: Man3GlcNAc2-PP-dolichol-alpha1,2-mannosyltransferase (to be further referred to as ALG11 mannosyltransferase), an ER-embedded protein that acts on a now nearly complete glycan moiety. ALG11 mannosyltransferase, encoded on the *ALG* gene on chromosome 13, catalyzes the cytosolic addition of the fourth and fifth alpha-1,2-linked mannose to alpha-D-Man-(1-3)- [alpha-D-Man-(1-6)]-beta-D-Man-(1-4)-beta-D-GlcNAc-(1-4)-alpha-D-GlcNAc-diphosphodolichol, also known as Man_3_GlcNAc_2_-PP-Dol, using the substrate GDP-mannose to produce alpha-D-Man-(1-2)-alpha-D-Man-(1-2)-alpha-D-Man-(1-3)- [alpha-D-Man-(1-6)]-beta-D-Man-(1-4)-beta-D-GlcNAc-(1-4)-alpha-D-GlcNAc-diphosphodolichol, also known as Man_5_GlcNAc_2_-PP-dol.

ALG11 mannosyltransferase produces the complete lipid-linked oligosaccharide in the cytosol. Marking the end of the cytosolic reactions, ALG11 mannosyltransferase catalysis is spatially and temporally linked to a subsequent flippase reaction that translocates the lipid-linked oligosaccharide into the lumen of the ER. Once in the ER, the moiety can be further processed and eventually added to nascent proteins at the amide nitrogen of asparagine (N) residues. Insults to the structure and/or enzymatic activity of ALG11 mannosyltransferase have implications on the N-glycosylation pathway and can lead to multisystemic pathology. ALG11-CDG, previously known as CDG-Ip, is a disorder stemming from a deficiency in ALG11 mannosyltransferase and manifests at an early age with severe symptoms [124] (Figure 1). Symptoms of ALG11-CDG impact neurological function and normal development, particularly through developmental and language delay, axial hypotonia, epileptic seizures, dysmorphic facial features, deafness, and inverted nipples [125,126] (Table 1). Neurological symptoms in ALG11-CDG are associated with the important role of ALG11 mannosyltransferase in the N-glycosylation pathway. Glycoproteins have been considered as critical components of the extracellular matrix which, in the nervous system [127], anchor nerve axons to allow for efficient neural communication and nerve development. Deficiencies in N-glycosylation may therefore compromise nervous system formation and function [128].

Like other CDGs, ALG11-CDG is diagnosed using TIEF to expose hypoglycosylation [95]. Recently, however, there have been studies that may have identified a potentially more sensitive biomarker of ALG11-CDG. Glycoprotein 130 (GP130) is the ubiquitously expressed cytokine receptor forming one subunit of the type I cytokine receptor within the IL-6 receptor family. It was shown to be hypoglycosylated and abnormally truncated in ALG11-CDG patients with normal transferrin carbohydrate composition [129]. Utilizing the GP130 biomarker may bolster the diagnostic toolset in ALG11-CDG diagnostic testing given the biomarker’s particular sensitivity in detecting hypoglycosylation despite persistent normal transferrin composition. Furthermore, whole exome sequencing analysis also allows for the molecular diagnosis of ALG11-CDG, helping to better understand the disorder itself and identify carriers of the disorder as well [95].

To date, there are no successful treatments for patients suffering from ALG-11 mannosyltransferase deficiency [124], and all current therapies are focused on relieving symptoms of the disorder, such as utilizing surgical intervention for strabismus, a disorder where one’s eyes are not properly aligned with one another and point in different direction, or regimens of anticonvulsants for epileptic seizures [130].

While the three CDGs discussed in this review are distinct from one another in enzymatic deficiency, each CDG causes a defect in the N-glycosylation pathway that clinically presents through pathophysiology rooted in protein hypoglycosylation. The N-glycosylation pathway is dependent on a multiplicity of enzymes, and despite our focus on enzymatic deficiencies in the formation of the lipid-linked oligosaccharide, there are many CDGs related to the processing and attachment of the lipid-linked oligosaccharide to nascent proteins in the ER. Inborn errors of mannose metabolism cause a spectrum of disease in CDG patients, with symptoms ranging from mild to severe, illustrating the complexity of CDGs. Delving further into the mechanisms underlying the pathophysiology in CDG patients may offer an improved understanding of how CDG pathophysiology arises, and how CDG patients can be treated in the clinic.

### 3.2. Mannose Metabolism and Other Diseases

Mannose can be taken up by cells through GLUTs and through the function of HK, and MPI can contribute to the carbon flux into central carbon metabolism. It is estimated that the physiological concentration of mannose is 40 µM [131]. Of note, a recent study has shown that the mannose uptake by a panel of cancer cells following millimolar mannose supplementation leads to the accumulation of M6P, which subsequently impairs glucose flux into central carbon metabolism and inhibits cancer growth [132]. In a similar vein, mannose supplementation was also shown to have an anticancer effect in non-small cell lung cancer (NSCLC) cells through inhibiting proliferation and increasing cell death [133]. Translating these recent findings into the antineoplastic role of mannose may lead to a novel route of treatment for cancer and improved outcomes for cancer patients.

In addition to the therapeutic effects on certain cancers, mannose has been shown to have a beneficial role as a prophylaxis for urinary tract infections (UTIs) in women. Daily mannose supplementation has been reported to significantly lower the incidence of recurrent UTI to levels comparable to Nitrofurantoin, an established antibiotic for UTI treatment [134]. A later study with different cohorts also showed that a prophylactic regimen of mannose supplementation reduced the recurrence of UTIs and significantly improved the quality of life for subjects [135]. Such beneficial effects may be associated with bacterial lectin (a glycan structure-binding protein) binding to urothelial mannosylated proteins as is seen in the case of uropathogenic *Escherichia coli* (UPEC) targeting highly mannosylated proteins during invasion of the urothelium. The interaction between FimH, a lectin located at the tip of bacterial pili, and high-mannose-content structures on the urothelial proteins is critical for the ability of UPEC to colonize and invade the urothelium. This interaction can be blocked by excessive mannose due to a higher affinity of mannose to FimH.

Mannose has also been shown to contribute to a beneficial gut microbiome composition and a reduction of the development of metabolic disease when used as a nutritional prophylaxis. In a mouse model, mannose supplementation initiated early in life was shown to lead to a lean phenotype (reduction of weight gain, fat mass, liver steatosis) and improve glucose tolerance of high-fat diet-fed mice [136], which is associated with mannose-mediated changes in gut microbial composition. While these benefits are yet to be seen in human trials, this finding elucidates an interesting contrast between metabolic outcomes associated with fructose and mannose. Such contrast would be linked to their different metabolic fates in central carbon metabolism as well as their distinct contribution to glycosylation. Fructose activates glycolysis and fatty acid synthesis by entering the pathway via KHK and ALDOB reaction whereas mannose impedes central carbon metabolism, which can reduce tumor growth. On the other hand, mannose is critical for protein and lipid glycosylation, which impacts cell–cell interaction and to ECM via lectin while fructose is not involved in glycosylation. Given that most surface proteins are glycosylated (e.g., receptors, transporters, CD proteins), alteration of the degree of glycosylation by dietary mannose supplementation, but not fructose, would potentially have positive antineoplastic and prophylactic effects in other contexts where cell–cell interaction and cell–extracellular protein interaction is involved.

## 4. Concluding Remarks

Although many studies on metabolic diseases focus on glucose, there are also substantial ailments that arise due to perturbations in mannose and fructose metabolism pathways. Thus, research on these overlooked monosaccharides has increased over the past decade to shine new light on fructose and mannose metabolism. Substantial progress has been made in understanding the biological mechanisms, physiological, and health consequences of defective fructose and mannose metabolism. The severity of inborn errors of fructose metabolism depends on whether metabolites affected by targeted enzymes can be catabolized by other pathways (if accumulated) or supplemented (if depleted). For example, while KHK deficiency results in an accumulation of fructose, the sugar can be metabolized through other pathways and does not necessarily result in a health-threatening clinical presentation. While essential fructosuria does not cause adverse health effects, a lack of functional ALDOB (causing HFI) or lack of functional FBP1 (causing FBPase deficiency) do manifest symptomatically. Treatment options for the inborn errors need to be more carefully examined. Dietary restriction of fructose sources is an effective treatment, but complete fructose restriction is quite challenging given that other sugars, such as sorbitol and sucrose, can be converted to fructose. Therefore, further research may suggest alternative treatment methods and identify healthy thresholds for fructose intake and long-term management of the disease. These same enzymes affected in inborn errors also experience metabolic alterations in cancer. The identification of fructose metabolism-related proteins as having both catalytic and signaling roles enables the development of treatment options targeting pathways that support cancer progression.

Mannose, closely related to fructose through the interconversion of F6P and M6P by MPI, is necessary for glycosylation. Inborn errors of mannose metabolism include MPI-CDG, PMM2-CDG, and ALG11-CDG in which patients experience symptoms as a consequence of improper protein glycosylation. Additionally, altered glycosylation is characteristic of aggressive cancers [137]. Thus, studies have considered mannose pathway enzymes like MPI in cancer to reveal an important role for balanced fructose and mannose metabolism. However, further research on how glycosylation is affected by other metabolic pathways and vice versa can bring clarity in identifying targets to prevent glycosylation associated with aggressive cancer phenotypes.

Understanding the roles of fructose and mannose across disease environments highlights the diversity of roles these molecules have and the multiple ways in which they support or suppress cell health. Knowing these intricacies of balanced sugar metabolism can determine which pathways are susceptible to therapeutic targeting.

## Figures and Tables

**Figure 1 metabolites-11-00479-f001:**
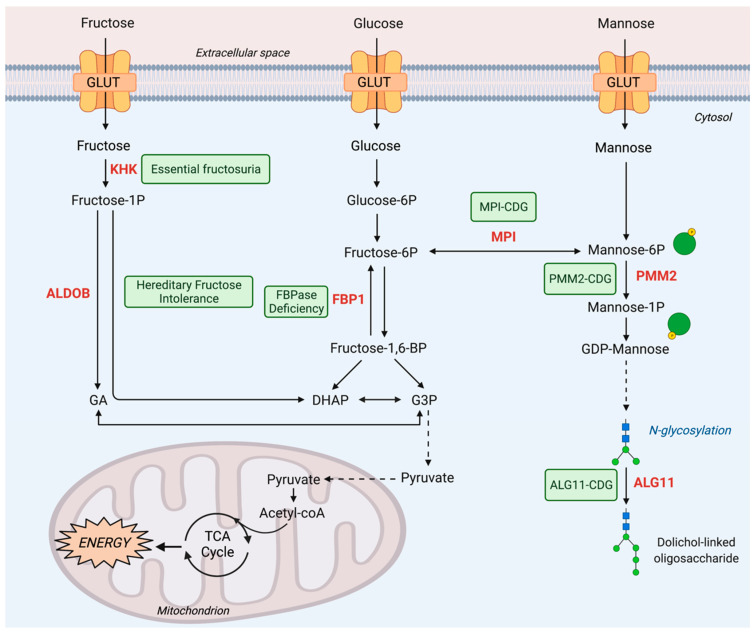
Fructose and mannose inborn errors of metabolic disease. A number of inborn errors are related to perturbed sugar metabolism, including fructose and mannose. Fructose and mannose transport occurs through glucose transporters due to their similarity in structure. Essential fructosuria and hereditary fructose intolerance affect fructolysis through deficiency in KHK and ALDOB, respectively. These pathways feed into the glycolysis pathways through GA to G3P conversion and DHAP production. FBPase deficiency affects fructose metabolism by loss of FBP1 activity blocking gluconeogenesis. Mannose metabolism, which is linked to fructose metabolism by the enzyme MPI, is also subject to mutations causing inborn errors. The central role of mannose in glycosylation explains the inborn errors classified as congenital disorders of glycosylation. MPI-CDG, PMM2-CDG, and ALG11-CDG relate to the deficiency of the enzyme listed in their names causing hypoglycosylation. These changes affect proper protein folding, leading to aberrant cellular function. Types of inborn errors of metabolism are shown in light green boxes. Proteins: GLUT, Glucose transporter; KHK, Ketohexokinase; ALDOB, Aldolase B; FBP1: Fructose-1,6-bisphosphatase 1; MPI: Mannose phosphate isomerase; PMM2: Phosphomannomutase 2; ALG11: GDP-Man:Man3GlcNAc2-PP-dolichol-alpha1,2-mannosyltransferase; Metabolites: GA; Glyceraldehyde; DHAP: Dihydroxyacetone phosphate; Fructose-1P: Fructose-1-phosphate; Glucose-6P: Glucose-6-phosphate; Fructose-6P: Fructose-6-phosphate; Fructose-1,6BP: Fructose-1,6-bisphosphate; Mannose-6P: Mannose-6-phosphate; Mannose-1P: Mannose-1-phosphate; GDP-mannose: Guanosine diphosphate mannose; G3P, Glyceraldehyde-3-phosphate; Pathways: TCA, tricarboxylic acid; Disease: CDG: Congenital disorders of glycosylation.

**Figure 2 metabolites-11-00479-f002:**
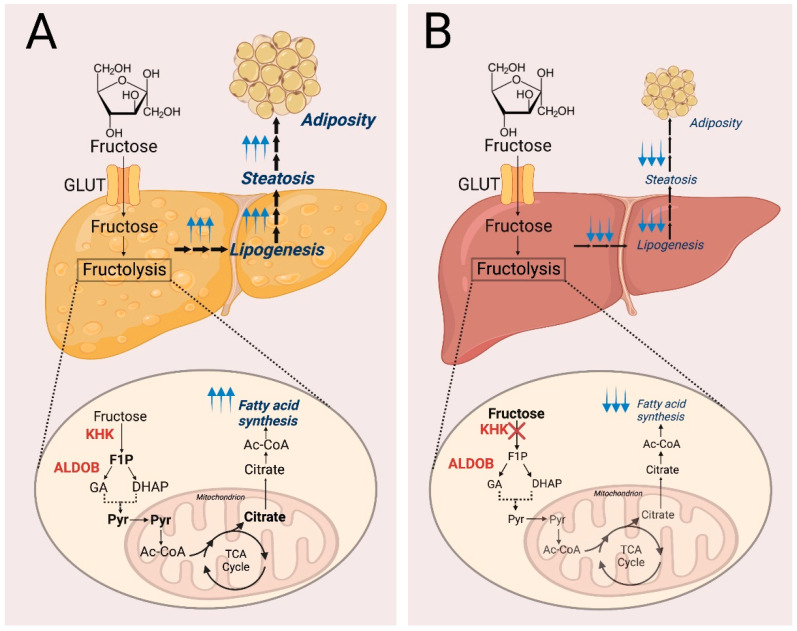
The protective role of KHK deletion against the development of sugar-induced metabolic syndrome. (**A**) Hepatocytes with functional KHK can metabolize fructose through fructolysis and this can contribute to liver steatosis and adiposity. (**B**) KHK deficiency in hepatocytes inhibits fructolysis and serves a protective role against liver steatosis and adiposity. Enzymes are shown in bold red, metabolites are black, pathway outcomes are italicized navy. Upwards and downwards blue arrows are indicative of activity being upregulated or downregulated, respectively. Proteins: GLUT, Glucose transporter; KHK, Ketohexokinase; ALDOB, Aldolase B; Metabolites: F1P, Fructose-1-phosphate; GA, Glyceraldehyde; DHAP, Dihydroxyacetone phosphate; Pyr, Pyruvate; Ac-CoA, Acetyl-CoA; Citrate; Pathway: TCA cycle, Tricarboxylic acid cycle.

**Figure 3 metabolites-11-00479-f003:**
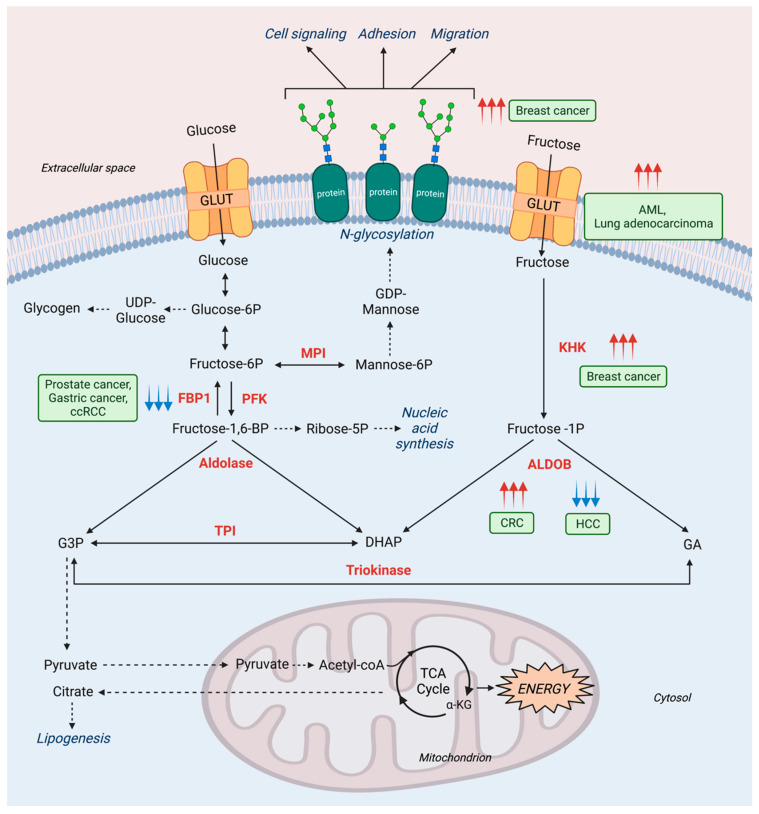
Fructose metabolism in cancer. Cancer cells reprogram their metabolism to support the greater energy demand for cell growth and proliferation. Often, fructose can be used as an alternative fuel source in low glucose conditions. The ability of fructose to be metabolized through various pathways (i.e., glycolysis, gluconeogenesis, glycosylation, nucleic acid synthesis, lipogenesis) paves the way for a number of metabolic alterations to occur in cancer, which are shown in the diagram above. The upregulation of cell membrane transporters, such as GLUT, induce increased fructose metabolism. Furthermore, the altered expression of enzymes, such as KHK, FBP1, and ALDOB, contribute to substantial clinical presentation of tumor progression in cancers, such as breast, prostate, gastric, ccRCC, and CRC. Enzymes are shown in bold red, metabolites are black, pathway outcomes are italicized navy. Cancer types are shown in light green text boxes. Red and blue arrows are indicative of expression or activity being upregulated or downregulated, respectively. Proteins: GLUT, Glucose transporter; FBP1, Fructose-1,6-bisphosphatase 1; PFK, Phosphofructokinase; TPI, Triose-phosphate isomerase; KHK, Ketohexokinase; ALDOB, Aldolase B; MPI: Mannose phosphate isomerase; Metabolites: αKG, Alpha-ketoglutarate; GA, Glyceraldehyde; DHAP, Dihydroxyacetone phosphate; Ribose-5P, Ribose-5-phosphate; Glucose-6P, Glucose-6-phosphate; UDP, Uridine diphosphate; Fructose-1P, Fructose-1-phosphate; Fructose-6P, Fructose-6-phosphate; Fructose-1,6-BP, Fructose-1,6-bisphosphate; Mannose-6P, Mannose-6-phosphate; GDP, Guanosine diphosphate; G3P, Glyceraldehyde-3-phosphate; Pathways: TCA, Tricarboxylic acid. Disease: CRC, Colorectal cancer; HCC, Hepatocellular carcinoma; ccRCC, Clear cell renal cell carcinoma; AML, Acute myeloid leukemia.

**Table 1 metabolites-11-00479-t001:** Clinical manifestation and treatment of inborn errors of fructose and mannose metabolism.

Metabolite	Disorder	Enzyme Deficiency	Affected Organs/Organ System	Clinical Manifestations	Treatment
**Fructose**	Essential Fructosuria	KHK	Liver, circulatory system	Fructosuria	No treatment necessary
	Hereditary Fructose Intolerance	ALDOB	Liver, kidney	Hypoglycemia, nausea, vomiting, abdominal pain upon fructose consumption, aminoaciduria, fructosuria	Dietary restriction of fructose, sorbitol, and sucrose
	FBPase Deficiency	FBP1	Liver, brain, heart	Severe hypoglycemia, hepatomegaly, acidoketosis, episodic acute crises of hyperventilation, coma, seizures, irritability, tachycardia	Dietary restriction of fructose, oral/IV glucose administration, avoidance of fasting, supplementing water with cornstarch to prevent nocturnal hypoglycemia
**Mannose**	MPI-CDG	MPI	Digestive tract	Liver fibrosis, hyperglycemia, protein-loss enteropathy, failure to thrive	Oral mannose
	PMM2-CDG	PMM2	Central nervous system, circulatory system	Infantile multisystem: developmental delays, failure to thrive, and neuropathy Late-infantile and childhood ataxia-intellectual disability: language and motor delays, stroke-like episodes Adult stable disability: premature aging, abnormal sexual development, and increased risk of deep-vein thrombosis	Occupational therapy, maintenance of healthy blood glucose
	ALG11-CDG	ALG11	Central nervous system, circulatory system, endocrine system	Developmental and language delay, axial hypotonia, seizures, dysmorphic facial features, inverted nipples	Plasma infusions, hormonal regimens

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
