# Peer review of "Fructose and Mannose in Inborn Errors of Metabolism and Cancer"

_metabolites, 2021, doi:10.3390/metabo11080479_

Round 1

Reviewer 1 Report

Although this manuscript is overall well-written, I'm finding it hard to understand what the authors want to transmit with this review paper. The majority of the information reported in this manuscript are well-known, -established and reported in all the biochemistry books. Each first-year university student already knows these concepts. Where is the novelty? The authors mention "metabolic diseases" in the title, but they talk about cancer in the text. Where is the evidence summarised that justifies the aim of this paper?

Author Response

We thank the three Referees for their detailed analyses and suggestions of the paper. The recommendations have been incorporated into the revised manuscript, and again we thank the Referees for their attention to the paper.

Each comment from all three Referees is addressed below.

Reviewer 1

Although this manuscript is overall well-written, I'm finding it hard to understand what the authors want to transmit with this review paper. The majority of the information reported in this manuscript are well-known, -established and reported in all the biochemistry books. Where is the novelty? The authors mention "metabolic diseases" in the title, but they talk about cancer in the text. Where is the evidence summarised that justifies the aim of this paper?

  1. Novelty issue:

While most information mentioned in the manuscript has been known, hardly any review article has covered both fructose and mannose metabolism. Especially, there has been no review paper to explain why consequences of perturbations in fructose metabolism and mannose metabolism on cancer are different. In this manuscript, we compared these two related, yet different metabolic pathways and their relationship with glycolysis, providing possible explanations of the contrast between fructose (oncometabolite) and mannose (tumor-suppressing metabolite) (line 660-670). 

By integrating both fructose and mannose metabolism and trying to understand the different roles of these two sugars in cancer metabolism, we believe the manuscript has its own novelty. Thus, our work not only integrates both fructose and mannose metabolism in two metabolic diseases (inborn error of metabolism and cancer), it provides a potential molecular mechanism by which fructose and mannose metabolism could cause the opposite consequence on cancer metabolism. We therefore believe that the review will have its own novelty and be of interest to the readership of Metabolites.

  1. Title change:

We agree that vast majority of this manuscript is related to inborn error metabolism and cancer. As per Referee’s concerns, we edited the title to ‘Fructose and Mannose in Inborn Error Diseases and Cancer’.

Reviewer 2 Report

The authors have well writing in this manuscript. They not only presented detail figures for the important pathways but also update recent critical knowledge. The only minor thing needs to be improved is that, metabolism of fructose and mannose involves the interaction of both liver and adipose tissue, the physiopathological changes of this part should be included. It will be appreciated to have a figure for this issue.

Author Response

We thank the three Referees for their detailed analyses and suggestions of the paper. The recommendations have been incorporated into the revised manuscript, and again we thank the Referees for their attention to the paper.

Each comment from all three Referees is addressed below.

Reviewer 2

The authors have well writing in this manuscript. They not only presented detail figures for the important pathways but also update recent critical knowledge. The only minor thing needs to be improved is that, metabolism of fructose and mannose involves the interaction of both liver and adipose tissue, the physiopathological changes of this part should be included. It will be appreciated to have a figure for this issue.

We thank the Referee’s suggestion and incorporated the figure accordingly (New Figure 2).  

Reviewer 3 Report

Lieu et al. discuss pathophysiology linked to perturbations in fructose and mannose metabolism, diagnostic tools and treatment options of the diseases. Overall, this review is well written and carries update information. This review is suitable for this special issue.

Strength and limitation:

Strength: An update review article covers studies on metabolic diseases including glucose, mannose, and fructose metabolism pathways. The authors also propose dietary changes or alternative therapeutic strategies might target fructose-related pathways to mitigate cancer progression.

Limitation: Inborn errors metabolism are rarely encountereed in clinical practice. The target readers are relatively few.

Only a few minor concerns:

  1. A few editing errors and typos: line 179, [26] [b]; line 237 SAM/SAH?; line 242, eGFR, full name please; line 299 deficiency ; Mutation; line 326 (IV);
  2. The reference styles are inconsistent. For example, different styles were seen between 134 and 135.

Author Response

We thank the three Referees for their detailed analyses and suggestions of the paper. The recommendations have been incorporated into the revised manuscript, and again we thank the Referees for their attention to the paper.

Each comment from all three Referees is addressed below.

Reviewer 3

Lieu et al. discuss pathophysiology linked to perturbations in fructose and mannose metabolism, diagnostic tools and treatment options of the diseases. Overall, this review is well written and carries update information. This review is suitable for this special issue.

Strength and limitation:

Strength: An update review article covers studies on metabolic diseases including glucose, mannose, and fructose metabolism pathways. The authors also propose dietary changes or alternative therapeutic strategies might target fructose-related pathways to mitigate cancer progression.

Limitation: Inborn errors metabolism are rarely encountered in clinical practice. The target readers are relatively few.

We appreciate the Referee’s concerns. Although rare, inborn errors of metabolism are the result of inherited genetic defects in metabolic enzymes that lead to chemical imbalances in children. Thus, understanding inborn errors of metabolism is critical for those in medical and pediatric training.

Only a few minor concerns:

  1. A few editing errors and typos: line 179, [26] [b]; line 237 SAM/SAH?; line 242, eGFR, full name please; line 299 deficiency ; Mutation; line 326 (IV);
  2. The reference styles are inconsistent. For example, different styles were seen between 134 and 135.

We thank the Referee’s comments and revised accordingly.

Round 2

Reviewer 1 Report

Authors addressed all the issues raised